# Chemically defined and xeno-free culture condition for human extended pluripotent stem cells

Bei Liu[1,6], Shi Chen[1,6], Yaxing Xu[1,6], Yulin Lyu[2,6], Jinlin Wang[1], Yuanyuan Du[1], Yongcheng Sun[3], Heming Liu[1], Haoying Zhou[1], Weifeng Lai[1], Anqi Xue[1], Ming Yin[4], Cheng Li [2], Yun Bai [5✉], Jun Xu [5✉] & Hongkui Deng [1✉]

Extended pluripotent stem (EPS) cells have shown great applicative potentials in generating synthetic embryos, directed differentiation and disease modeling. However, the lack of a xeno-free culture condition has significantly limited their applications. Here, we report a chemically defined and xeno-free culture system for culturing and deriving human EPS cells in vitro. Xeno-free human EPS cells can be long-term and genetically stably maintained in vitro, as well as preserve their embryonic and extraembryonic developmental potentials. Furthermore, the xeno-free culturing system also permits efficient derivation of human EPS cells from human fibroblast through reprogramming. Our study could have broad utility in future applications of human EPS cells in biomedicine.

[1] Department of Cell Biology, School of Basic Medical Sciences, Peking University Stem Cell Research Center, State Key Laboratory of Natural and Biomimetic Drugs, Peking University Health Science Center and the MOE Key Laboratory of Cell Proliferation and Differentiation, College of Life Sciences, Peking-Tsinghua Center for Life Sciences, Peking University, Beijing, China. [2] School of Life Sciences, Center for Bioinformatics, Center for Statistical Science, Peking University, Beijing, China. [3] Bengbu Medical College, Bengbu, Anhui, China. [4] Beijing Vitalstar Biotechnology, Beijing, China. [5] Department of Cell Biology, School of Basic Medical Sciences, Peking University Stem Cell Research Center, Peking University Health Science Center, Peking University, Beijing, China. [6] These authors contributed equally: Bei Liu, Shi Chen, Yaxing Xu, Yulin Lyu. ✉email: baiyun@bjmu.edu.cn; jun_xu@bjmu.edu.cn; hongkui_deng@pku.edu.cn

Pluripotent stem cells (PSCs) can be generated from embryos and somatic cells[1–4], which are invaluable for studying developmental biology, disease modeling and regenerative applications. However, conventional PSCs rarely contribute to trophectoderm-derived lineages in vivo[5]. This limitation has been overcome by establishing stem cells that can generate both embryonic and extraembryonic lineages in vivo[6–9]. These stem cells, which are designated as either extended pluripotent stem (EPS) cells[6] or expanded potential stem (EPS) cells[8], provide new tools to study the regulation of developmental potency.

In addition to their expanded developmental potentials, EPS cells also have a superior chimeric ability and germline-competence. A single mouse EPS cell permits generating single-EPS-cell-derived mice by tetraploid complementation[6], which has been applied to directly generate gene-targeted mouse models[10,11]. In addition to mouse EPS cells, human EPS (hEPS) cells show interspecies chimeric capacity in mouse conceptuses[6]. Notably, a very recent application of EPS cells has allowed the generation of blastocyst-like structures in vitro[12,13], which provides valuable platforms for exploring early embryogenesis in vitro. These studies have highlighted the promising applicative advantages of EPS cells in basic and translational research.

Most recently, our group has demonstrated that hEPS cells can generate functional human hepatocytes via directed differentiation[14]. Notably, CellNet analysis showed that their gene regulation network highly resembled that of primary human hepatocytes, and overcame with that of hepatocytes from conventional human PSCs. Methodologically, our step-wise protocol included the pretreatment of hEPS cells into a primed pluripotent-like state, which was followed by a hepatic differentiation protocol for conventional human PSCs. In principle, this strategy could be generally applied to generate other lineages from hEPS cells by simply adapting differentiation protocols for conventional human PSCs. In this regard, hEPS cells are promising in generating a wide range of functional cell types for therapeutics. However, the therapeutic translation of PSC-derived cell types requires xeno-free culturing of PSCs[15], which remains unexplored. In this study, after systematically optimizing the original LCDM condition, we establish a chemically defined and xeno-free culture system that can support derivation and long-term stable propagation of hEPS cells in vitro.

## Results

**Establishment of a xeno-free culturing condition for human EPS cells.** To solve this problem, we first identified feeder cell-secreted factors that are critical for maintaining self-renewal of hEPS cells. Using an OCT4-Tdtomato hEPS cell line (Supplementary Fig. 1a), we found that Activin A efficiently maintained more than 80% of OCT4 + hEPS cells (Fig. 1a and Supplementary Fig. 1b), and the expression of key pluripotent marker genes in feeder-free hEPS cells (Supplementary Fig. 1c). Next, we further optimized the feeder-free culturing condition by adopting a simplified LCDM medium that contained components of ITSX (insulin, transferrin, sodium selenite and ethanolamine) to replace N2 and B27. Basing on this simplified LCDM medium, we found that the combination of catalase and vitamin C further promoted hEPS cell proliferation (Fig. 1b and Supplementary Fig. 1d). We also found that a 1:1 mixture of DF12 and Neurobasal medium was optimal for their proliferation (Fig. 1c).

Next, we identified the matrix proteins that could replace Matrigel (which varies from batch to batch and contains unknown animal components). We found that Laminin 521 consistently promoted the attachment, survival and proliferation of hEPS cells (Fig. 1d–f and Supplementary Fig. 2a, b). Under this optimized condition, hEPS cells with different genetic backgrounds could be propagated in vitro (Fig. 1g). Furthermore, the absence of non-human mammalian components was suggested by the negative expression of Neu5Gc (Supplementary Fig. 2c). Collectively, these results indicate that we established a chemically defined and xeno-free culture condition for culturing hEPS cells with different genetic backgrounds (Supplementary Tables 1,2).

**Xeno-free human EPS cells can be genetically stably maintained after long-term propagation.** To characterize hEPS cells cultured in the xeno-free LCDM medium (xeno-free hEPS cells), we first analyzed their proliferative ability. The doubling time of xeno-free hEPS cells was ~15 h (Fig. 2a), which was consistent with the presence of 50% EdU+ replicating hEPS cells (Fig. 2b and Supplementary Fig. 3a). We next found the single-cell cloning efficiency of xeno-free hEPS cells to be more than 50% (Fig. 2c). Notably, a low dosage of Rock-inhibitor treatment was important for single-cell survival and proliferation of xeno-free hEPS cells (Supplementary Fig. 3b, c). Additionally, xeno-free hEPS cells could be passaged by using different dissociation reagents (Supplementary Fig. 3d). Importantly, the rapid proliferation and efficient single-cell cloning of xeno-free hEPS cells allowed successful gene targeting in these cells (Supplementary Fig. 3e).

We further analyzed the genetic stability of xeno-free hEPS cells after long-term culturing. Karyotype analysis showed that xeno-free hEPS cells with different genetic backgrounds could maintain normal karyotype after more than 20 passages (Fig. 2d). Consistent with this, the intensity of rH2AX staining signals, an indicator of genotoxic effects, were at comparable levels between xeno-free hEPS cells and hEPS cells cultured on feeders (Supplementary Fig. 3f). We also performed whole-genome sequencing on these cells, and found that the frequencies of genetic indels after long-term passaging were at comparable levels between xeno-free and feeder-cultured hEPS cells (Fig. 2e and Supplementary Data 1). These results indicated that xeno-free hEPS cells could be genetically stably maintained long-term in vitro.

**Characterization of the molecular features of xeno-free human EPS cells.** Next, we analyzed the key molecular features of extended pluripotency in xeno-free hEPS cells, and confirmed the expression of key pluripotency markers (Fig. 3a and Supplementary Fig. 4a, b). In addition, enhanced OCT4-distant enhancer activity was also maintained in xeno-free hEPS cells (Supplementary Fig. 4c). Moreover, female xeno-free hEPS cells exhibited a range of different X-chromosome inactivation (XCI) states with some cells showing no XIST, monoallelic XIST, or biallelic XIST expression (Supplementary Fig. 4d). We further analyzed the global transcriptome of xeno-free hEPS cells, and found that these cells had a global gene expression profile similar to hEPS cells cultured on feeders (Supplementary Fig. 5a), which is consistent with the result of clustering analysis (Fig. 3b). We also identified 169 genes that were differently expressed between xeno-free and feeder-cultured hEPS cells (Supplementary Fig. 5b), the function of which were majorly related to catabolic process of metabolites and RNA polymerase-mediated transcriptional regulation. We further found that xeno-free hEPS cells maintained expression of core pluripotency genes and upregulated several naive pluripotency markers, such as DPPA3, KLF17 and KLF4, when compared to primed human PSCs (hPSCs) (Supplementary Fig. 5c). Collectively, these data suggested that xeno-free hEPS cells maintained the molecular features of extended pluripotency.

In our previous study[6], we identified two gene modules that are specifically enriched in hEPS cells, which were involved in

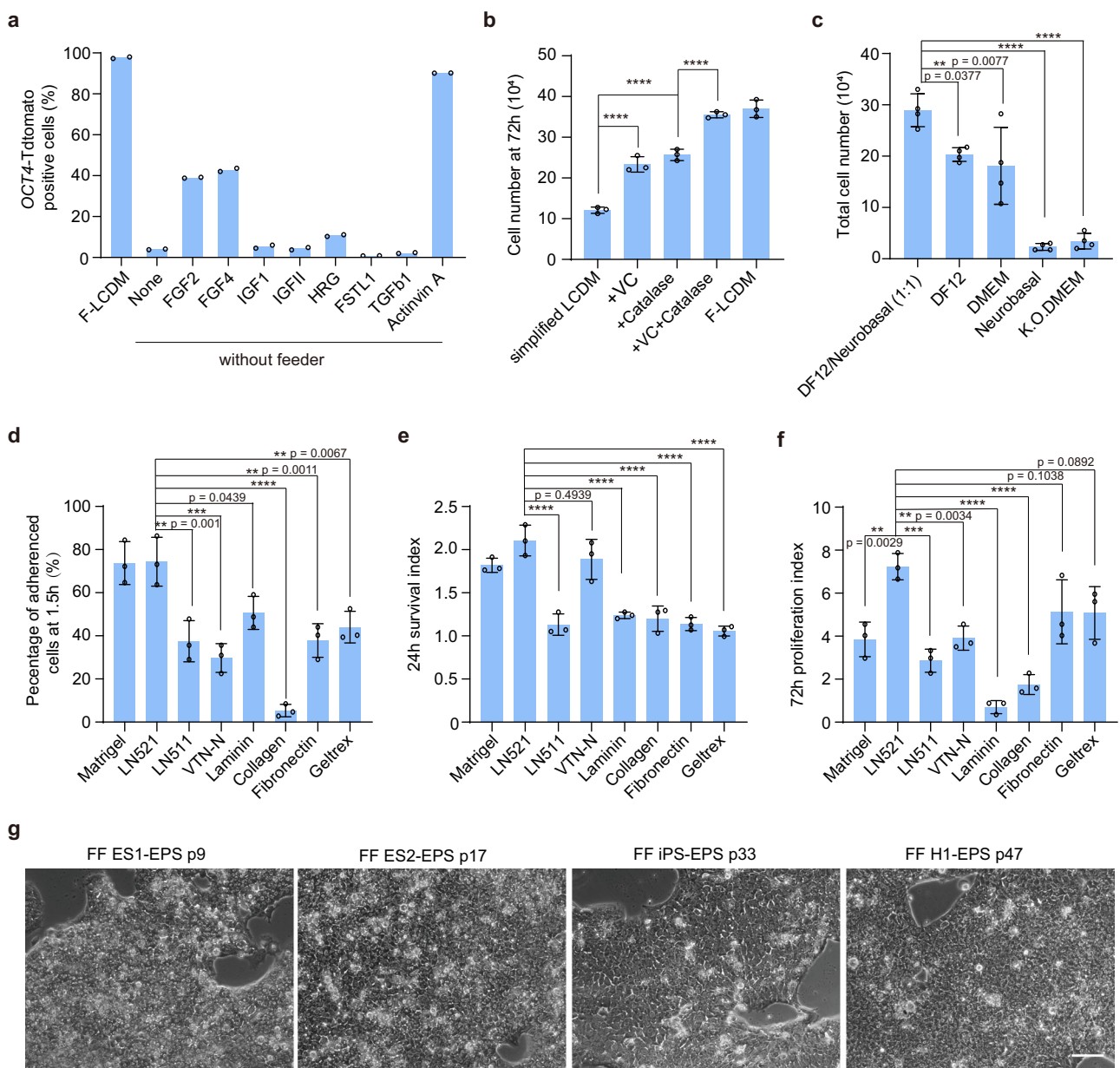

**Fig. 1 Development of a xeno-free culturing condition for human EPS cells. a** FACS analysis of the percentages of OCT4-Tdtomato+ H1-EPS cells with indicated factor treatment after 2 passages. $n = 2$ biologically independent samples. None, LCDM medium. **b** Analysis of cell numbers of feeder-free cultured hEPS cells under the treatments of vitamin C, catalase, and their combination. VC, vitamin C. $n = 3$ biologically independent samples. ES1-EPS cells were used. **c** Analysis of cell numbers of feeder-free cultured hEPS cells in different basal culturing media. $n = 4$ biologically independent samples. ES1-EPS cells were used. **d–f** Analysis of the effects of different matrix proteins on feeder-free cultured hEPS cells. The percentages of adherent hEPS cells at 1.5 h after seeding was shown in **d**. Survival of dissociated hEPS cells 24 h after seeding was shown in **e**. Proliferation of hEPS cells at 72 h after seeding was shown in **f**. For **e**, **f**, index represents the cell number at a specific time point divided by the number of seeding cells. For **d–f**, $n = 3$ biologically independent samples. ES1-EPS cells were used. **g** Representative morphologies of xeno-free hEPS cells with different genetic backgrounds. Similar images were obtained in at least 5 independent experiments. Error bars, mean ± SD. All differences between means with $P < 0.01$ are indicated. **$P < 0.01$; ***$P < 0.001$; ****$P < 0.0001$. Statistical significance was analyzed using one-way ANOVA with Tukey multiple comparison test in **b–f**. Scale bar, 100 μm. Experiments in **a–g** were all independently repeated at least three times with similar results.

chromatin organization and transcriptional regulation (Module C) or associated with transcriptional features of zygote to 4 cell embryos (Module D). Notably, these two gene modules were also presented in xeno-free hEPS cells (Fig. 3c). Interestingly, we found that Module D was also enriched in trophectodermal lineage in blastocyst (Fig. 3c), which may be related to the trophoblast developmental potentials of hEPS cells. Next, we explored the transcriptional relationship between xeno-free hEPS cells and epiblast cells from human embryos using bulk and single-cell RNA seq data (Fig. 3d and Supplementary Fig. 5d)[16–21]. Xeno-free hEPS cells showed some transcriptional similarities to E5-E7 epiblast cells, and upregulation of naive transcriptional signatures (Fig. 3d and Supplementary Fig. 5d). When comparing with naive hPSCs, it is also notable that xeno-free and feeder-dependent hEPS cells still showed some transcriptional similarities to epiblast cells from post-implantation stages (Supplementary Fig. 5d).

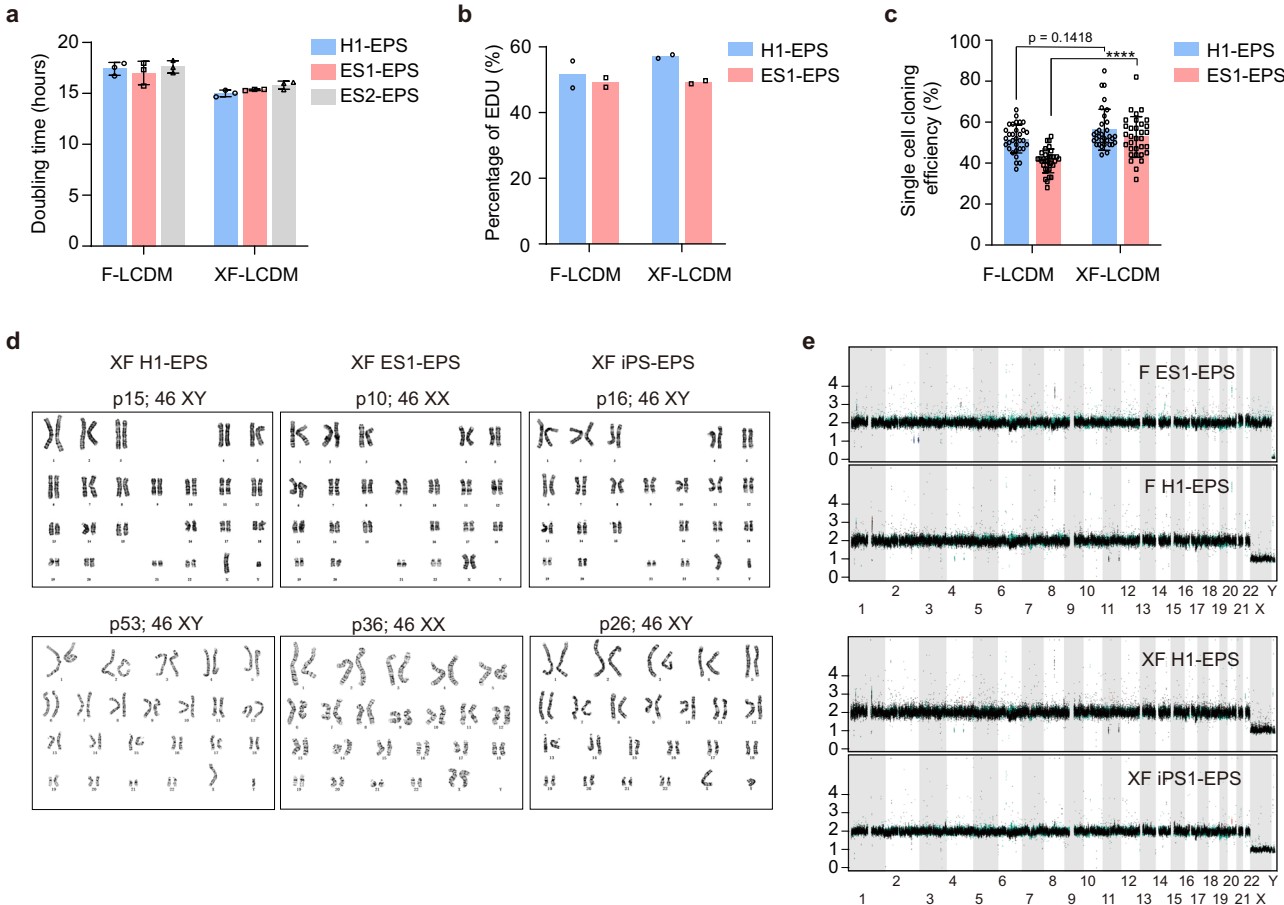

**Fig. 2 Xeno-free human EPS cells proliferate well and maintain genome stability after long-term culturing. a** Analysis of doubling time of feeder-cultured and xeno-free hEPS cells. $n = 3$ biologically independent samples. **b** FACS analysis of the percentages of EDU + cells in feeder-cultured and xeno-free hEPS cells. $n = 2$ biologically independent samples. **c** Single-cell cloning efficiency of feeder-cultured and xeno-free hEPS cells. $n = 32$ biologically independent samples. **d** Karyotype analysis of xeno-free hEPS cells with different genetic backgrounds. **e** Chromosomal copy-number analysis of EPS cells by whole-genome sequencing. F-LCDM, hEPS cells cultured on feeders; XF-LCDM, xeno-free hEPS cells. Error bars, mean ± SD. All differences between means with $P < 0.01$ are indicated. **$P < 0.01$; ***$P < 0.001$; ****$P < 0.0001$. Statistical significance was analyzed using one-way ANOVA with Tukey multiple comparison test (**a, c**). Experiments in **a–c** were all independently repeated at least three times with similar results.

Interestingly, analysis of the single-cell sequencing data revealed that there were no obvious heterogenous expression of pre- or post-implantation epiblast markers in both xeno-free and feeder-dependent hEPS cells (Supplementary Fig. 5e). Taken together, these data suggested that xeno-free hEPS cells maintain the gene module associated with early human embryos, and represent a unique state that is distinct from naive and primed hPSCs.

Next, we performed ATAC-seq analysis to explore chromatin accessibility in xeno-free hEPS cells. Compared to primed hPSCs or feeder-cultured hEPS cells, xeno-free hEPS cells showed more cell type-specific ATAC-seq peaks (Supplementary Fig. 5f). Using primed hPSCs as the control, we further analyzed function of genes related to cell type-specific ATAC-seq peaks in xeno-free and feeder-cultured hEPS cells using GO term analysis. Notably, most of the top GO terms in feeder-cultured hEPS cells could also be found in xeno-free hEPS cells, such as stem cell population maintenance, cell cycle G1/S phase transition, regulation of chromosome organization, and in utero embryonic development (Supplementary Data 2). It is also notable that unique top GO terms, such as viral gene expression and ncRNA processing, were presented in xeno-free hEPS cells (Supplementary Data 2). We further identified potential motifs for transcription factors in xeno-free and feeder-cultured hEPS cells (Supplementary Data 3). Generally, the identified motifs were similar between xeno-free

and feeder-cultured hEPS cells when compared to primed hPSCs (Supplementary Data 3), suggesting shared regulatory network of hEPS cells cultured in these two conditions. Motifs for key pluripotency genes such as OCT4-SOX2-TCF-NANOG were observed in both conditions. Interestingly, we also noticed that the chromatin landscape of genes involved in trophectoderm development was more open in xeno-free hEPS cells when compared to that in primed hPSCs (Fig. 3e). Collectively, these data suggested that xeno-free hEPS cells have shared chromatin accessibility landscape with feeder-cultured hEPS cells.

**Xeno-free human EPS cells showed embryonic and extra-embryonic developmental potentials.** To analyze the developmental potentials of xeno-free hEPS cells, we first performed in vitro embryoid body (EB) formation assays. Xeno-free hEPS cells showed robust differentiation potentials towards lineages of the three germ layers (Supplementary Fig. 6a), which was consistent with the results of teratoma formation in vivo (Fig. 4a and Supplementary Fig. 6b). We next characterized the extraembryonic differentiation potentials of xeno-free hEPS cells in vitro, and found that xeno-free hEPS cells could be induced into trophoblast-like cells using the BMP4-exposure-based protocol[22,23], as revealed by FACS analysis, Q-PCR and immunofluorescence (Supplementary Fig. 6c–f). Moreover, we

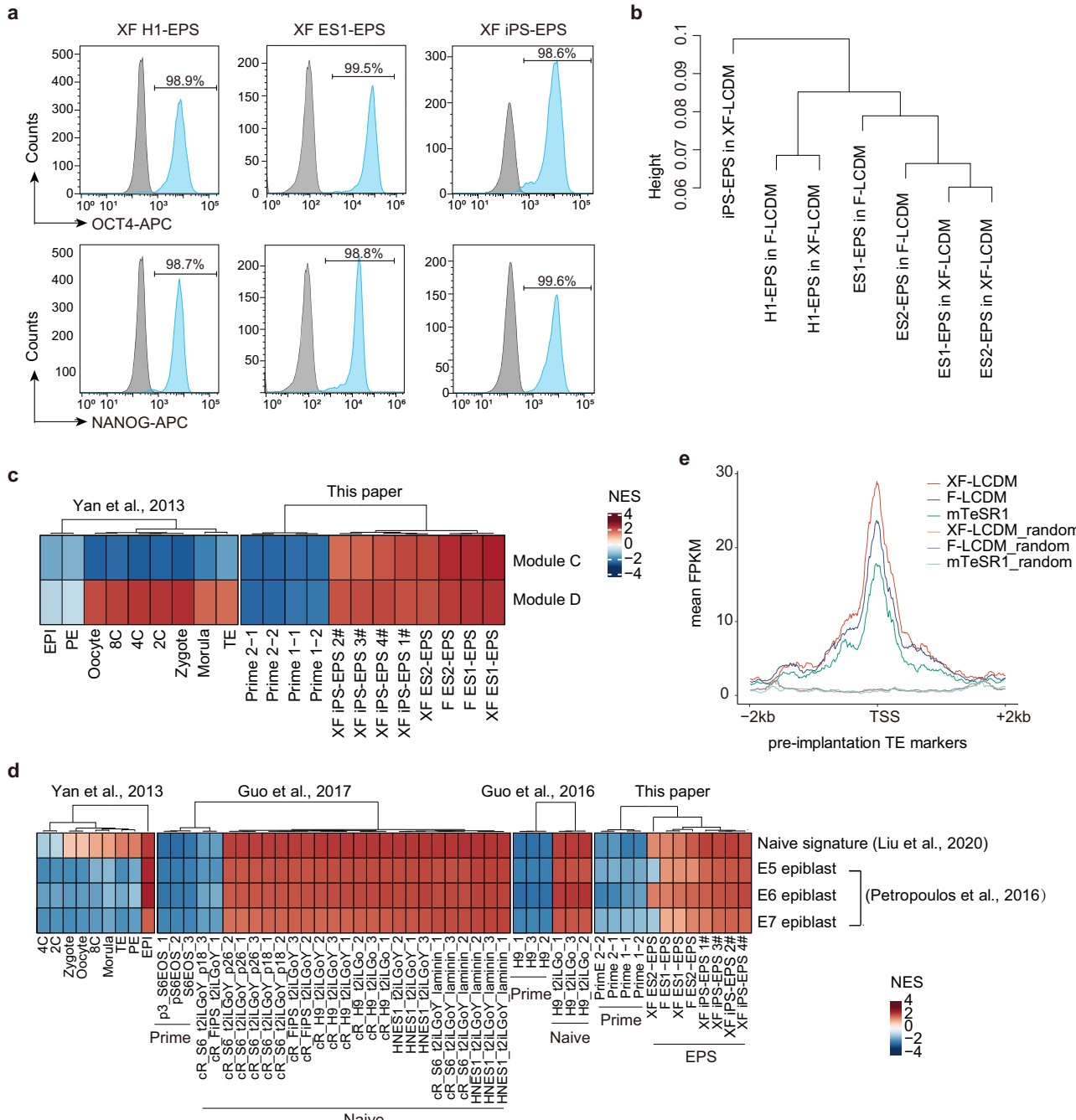

**Fig. 3 Characterization of molecular features of xeno-free human EPS cells. a** Representative FACS analysis of the percentages of OCT4 + or NANOG + cells in xeno-free hEPS cells. **b** Hierarchical clustering of global gene expression of EPS cells cultured in XF-LCDM and F-LCDM. Distance is calculated by 1- Spearman correlation coefficient. **c** GSEA analysis of enrichment of module C and module D in xeno-free and feeder-cultured hEPS cells. Bulk RNA-seq data for xeno-free and feeder-cultured hEPS cells were used. Gene signatures for module C and module D were defined in our previous study (Yang et al.)[6]. As the control, dataset for pre-implantation human embryos (GSE36552, Yan et al.)[20] were chosen. **d** GSEA analysis of enrichment of gene signatures for E5/ E6/E7 epiblasts or naive pluripotent stem cells in xeno-free and feeder-cultured hEPS cells. Bulk RNA-seq data for xeno-free and feeder-cultured hEPS cells were used. Gene signatures for E5/E6/E7 epiblasts were defined in the study performed by Petropoulos et al.[19], and naive signatures were defined in the study performed by Liu et al.[35]. As the control, dataset for pre-implantation human embryos (GSE36552, Yan et al.)[20], and naive or primed hPSCs (E-MTAB-4461, Guo et al.; E-MTAB-5674, Guo et al.)[17, 21] were chosen. **e** Mean FPKM values at TE marker genes in XF-LCDM, F-LCDM and mTeSR1 cultured primed cells. The signal density of given region (2 kb up- and down- streams of TSS) and randomly selected genome regions of the same length were analyzed. TE marker genes were defined in the study performed by Petropoulos et al.[19].

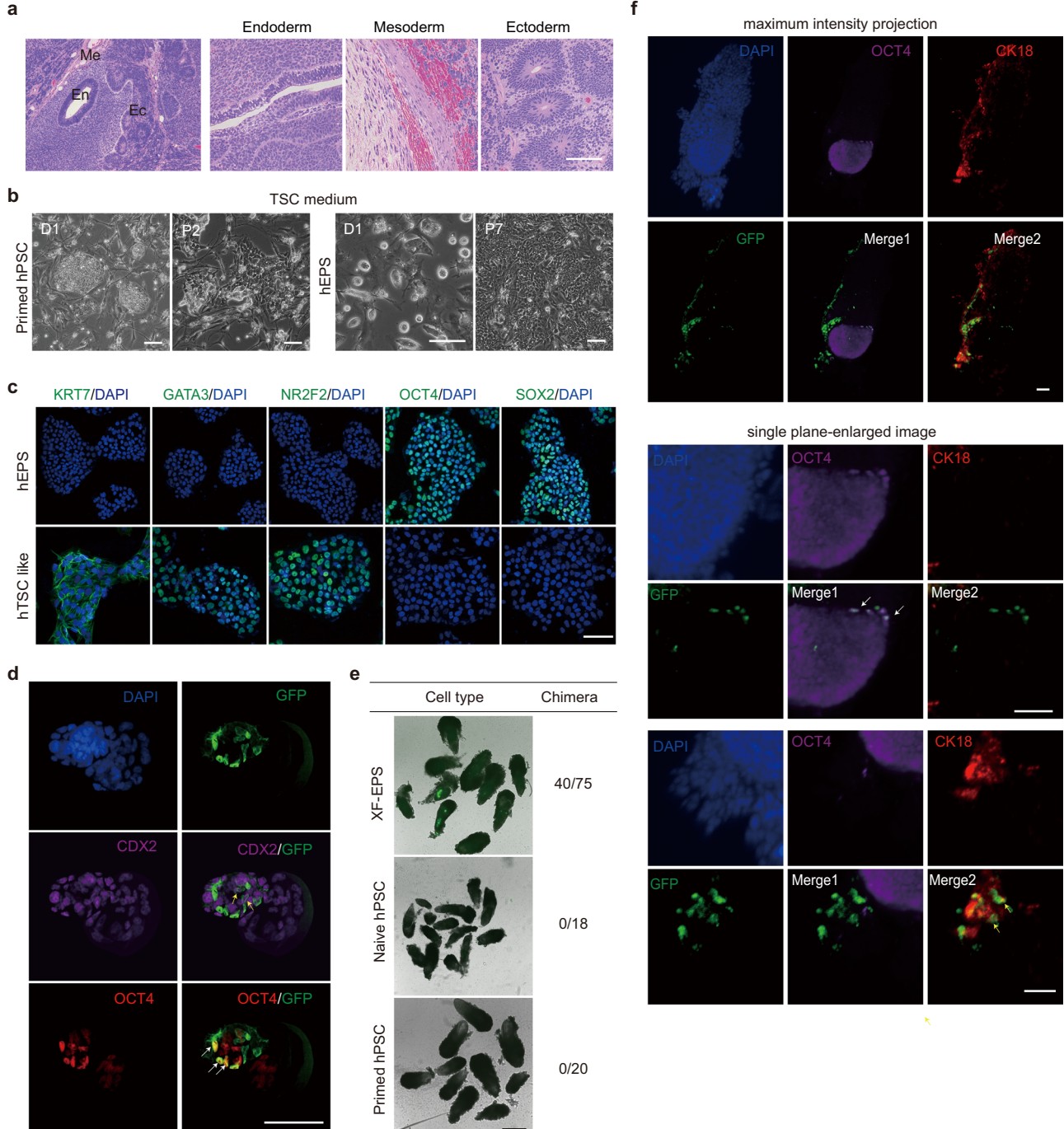

**Fig. 4 Characterization of embryonic and extraembryonic developmental potentials of xeno-free hEPS cells in vitro and in vivo. a** Representative images showing histological analysis of teratomas generated from xeno-free hEPS cells. Scale bars, 100 μm. Similar images were obtained in at least 3 independent experiments. **b** Phase contrast images of primed and XF-EPS cells. To derive hTSC-like cells, primed hPSCs and XF-EPS cells were cultured in hTSC medium. Primed H1 and XF H1-EPS cells were used. Scale bars, 100 μm. Similar images were obtained in at least three independent experiments. **c** Representative immune-staining images showing the marker expression of XF-EPS cells and hTSC-like cells. XF H1-EPS cells were used. Scale bars, 100 μm. Similar images were obtained in at least three independent experiments. **d** Representative images showing chimerism of xeno-free hEPS cells in E4.5 mouse blastocysts. GFP, anti-GFP antibody. White and yellow arrows indicate GFP + cells with OCT4 and CDX2 expression respectively. Scale bar, 100 μm. Similar images were obtained in at least 5 independent experiments. **e** Chimerism of XF-EPS cells, naive hPSCs and primed hPSCs in E6.5 mouse conceptuses. Representative images of chimerism were shown. A summary of the chimeric experiment was also shown. Scale bar, 500 μm. **f** Immunofluorescent staining of E6.5 chimeric embryos showing lineage contributions of xeno-free hEPS derivatives to epiblast or trophoblast lineages. mclover+ ES1-EPS cells were used. Embryos were stained for OCT4 (epiblast), CK18 (trophoblast compartment) and mclover (GFP). White and yellow arrows indicate GFP + cells with OCT4 and CK18 expression respectively. Scale bars, 50 μm. Similar images were obtained in at least 5 independent experiments. Experiments in **a–f** were all independently repeated at least three times with similar results.

also found that xeno-free hEPS cells could be converted into trophoblast stem-like cells after passaging in human trophoblast stem cell culturing medium[24] (Fig. 4b, c and Supplementary Fig. 6g), whereas primed hPSCs gradually died under such condition (Fig. 4b). Collectively, these data suggested that xeno-free hEPS cells can generate embryonic and extraembryonic lineages in vitro.

Next, we focused on evaluating the developmental potentials of xeno-free hEPS cells in vivo. First, xeno-free hEPS cells labeled with mclover reporter were injected into mouse 8-cell embryos, which were further cultured in the mouse embryo medium for 48 h in vitro. Analysis of the E4.5 injected mouse embryos showed that about 50% of analyzed embryos showed integration of hEPS derivatives in both inner cell mass and trophectoderm (Fig. 4d and Supplementary Fig. 7a, b). We further transplanted the chimeric E4.5 embryos into pseudo- pregnant mice, and found that 53.3% of retrieved E6.5 embryos (40/75) showed the presence of hEPS derivatives (Fig. 4e and Supplementary Fig. 7c, d). As the control, the in vivo E6.5 chimeric efficiency of naive (0/18) or primed (0/20) hPSCs was low (Fig. 4e). In E6.5 embryos, xeno-free hEPS derivatives showed the expression of marker genes for epiblast, trophoblast or extraembryonic endoderm (Fig. 4f and Supplementary Fig. 7e, f). We further confirmed the presence of derivatives of xeno-free hEPS cells in E10.5 mouse conceptuses by Q-PCR analysis of human-specific mitochondrial DNA element (Supplementary Fig. 7g). However, the chimeric efficiency of xeno-free hEPS cells (3.9%) in E10.5 mouse conceptuses was lower than that of feeder-cultured hEPS cells from our previous report[6]. Taken together, these data suggest that xeno-free hEPS cells have embryonic and extraembryonic developmental potentials in E6.5 mouse embryos.

**Derivation of xeno-free human EPS cells from primary human fibroblasts.** We next attempted to generate hEPS cells from human fibroblasts by reprogramming basing on our xeno-free LCDM medium (Fig. 5a). Xeno-free hEPS medium was used during the reprogramming process, and 5% of xeno-free knock-out serum replacement was also used to promote cell proliferation. After 10–14 days, emerged hEPS-like colonies could be further propagated in the xeno-free hEPS medium (Fig. 5b). We also confirmed the co-expression of pluripotent markers OCT4 and SOX2 in picked hEPS cell colonies (Supplementary Fig. 8a). hEPS cell lines could be established from picked colonies and single seeded cells (Supplementary Fig. 8b). hEPS cells derived from fibroblasts morphologically resembled those derived from embryos (Fig. 5c), retained expression of pluripotent marker genes as well as differentiation potentials (Fig. 5d, e and Supplementary Fig. 8c, d). Furthermore, they could be maintained long-term in vitro with normal karyotype (Fig. 5f). In addition, the global gene expression profiles of hEPS cells derived from fibroblasts were similar to hEPS cells cultured on feeders (Fig. 5g). Collectively, these results suggested that hEPS cells could be induced from somatic cells using our xeno-free EPS culturing condition.

## Discussion

In this study, we established a chemically defined and xeno-free culture system for maintaining and deriving hEPS cells. Xeno-free hEPS cells could be long-term maintained in vitro, and preserve the key molecular and functional features of extended pluripotency. Furthermore, our culture system also permitted generating xeno-free hEPS cells from somatic cells by reprogramming. In this regard, our xeno-free culture system would greatly promote the applications of hEPS cells.

Notably, the bi-differentiation potentials of xeno-free hEPS cells were demonstrated both in vitro and in vivo. In addition to

spontaneous EB differentiation to embryonic lineages, we also showed that xeno-free hEPS cells can be converted into human trophoblast stem-like cells using culturing medium for human trophoblast stem cells, whereas primed hPSCs gradually died under the same condition (Fig. 4b). This observation was consistent with one recent report showing successful conversion of feeder-cultured hEPS cells into human trophoblast stem cells in vitro[25]. The expanded developmental potentials of xeno-free hEPS cells were further supported by their contribution to both embryonic and extraembryonic lineages in E4.5 and E6.5 mouse embryos (Fig. 4d–f and Supplementary Fig. 7a–f). However, it is also important that cautions are needed for evaluating human-mouse interspecific experiments due to the limited chimeric efficiency, and future studies are required to enhance the interspecific chimeric ability of xeno-free hEPS cells.

Another important feature of xeno-free hEPS cells is their unique molecular characteristics that are distinct from naive and primed hPSCs. Similar to feeder-cultured hEPS cells, xeno-free hEPS cells showed the presence of several classical molecular features of naive pluripotency, including enhanced OCT4-distant enhancer activity and upregulation of naive transcriptomic features (Supplementary Fig. 4d–e). In addition, further transcriptomic analysis revealed transcriptomic resemblance of xeno-free hEPS cells to human epiblast cells from both pre- and post-implantation stages (Fig. 3d and Supplementary Fig. 5d). It is also interesting that no obvious heterogenous expression of pre- or post-implantation epiblast markers were observed in hEPS cell populations at the single-cell level (Supplementary Fig. 5e). More importantly, similar to feeder-cultured hEPS cells, the gene module associated with early human embryos was also presented in xeno-free hEPS cells (Fig. 3c). Notably, this module was also enriched in trophectoderm from pre-implantation human embryos (Fig. 3c), which is consistent with the observation that the chromatin landscape of genes involved in trophectoderm development was more open in xeno-free hEPS cells (Fig. 3e). Although these findings reveal transcriptional similarity of hEPS cells to early embryonic cells, future studies are required to unravel the exact developmental identity of these cells. It is possible that different subpopulations in hEPS cell cultures may resemble embryonic cells from different stages. Another possibility is that hEPS cells may represent a transient intermediate stage of human epiblast during the transition from pre- to post-implantation stages. Single-cell multi-omics analyses are required to clarify these possibilities in the future. Collectively, the findings regarding the molecular features of xeno-free hEPS cells could provide important clues for identifying the mechanisms regulating extraembryonic differentiation potentials of these cells in the future.

In summary, we have established a chemically defined and xeno-free culture condition for long-term maintaining and deriving hEPS cells, which could preserve the bi-differentiation potentials of hEPS cells. More importantly, the karyotype of xeno-free hEPS cells remains stable after long-term culturing in vitro, which is comparable to that of conventional feeder-free cultured hPSCs[26–28]. The maintenance of the gene module enriched in early human embryos and the more opened chromatin landscape in extraembryonic genes in xeno-free hEPS cells may be related to the their bi-differentiation potentials. Our culture system has promising potentials for the wide applications of hEPS cells in directed differentiation, disease modeling and developmental biology, and paves the way for translating EPS technology to the clinic.

## Methods

**Ethics statement and human sample collection.** Human embryonic fibroblasts were isolated from 2- to 3-month-old embryos that were obtained with informed

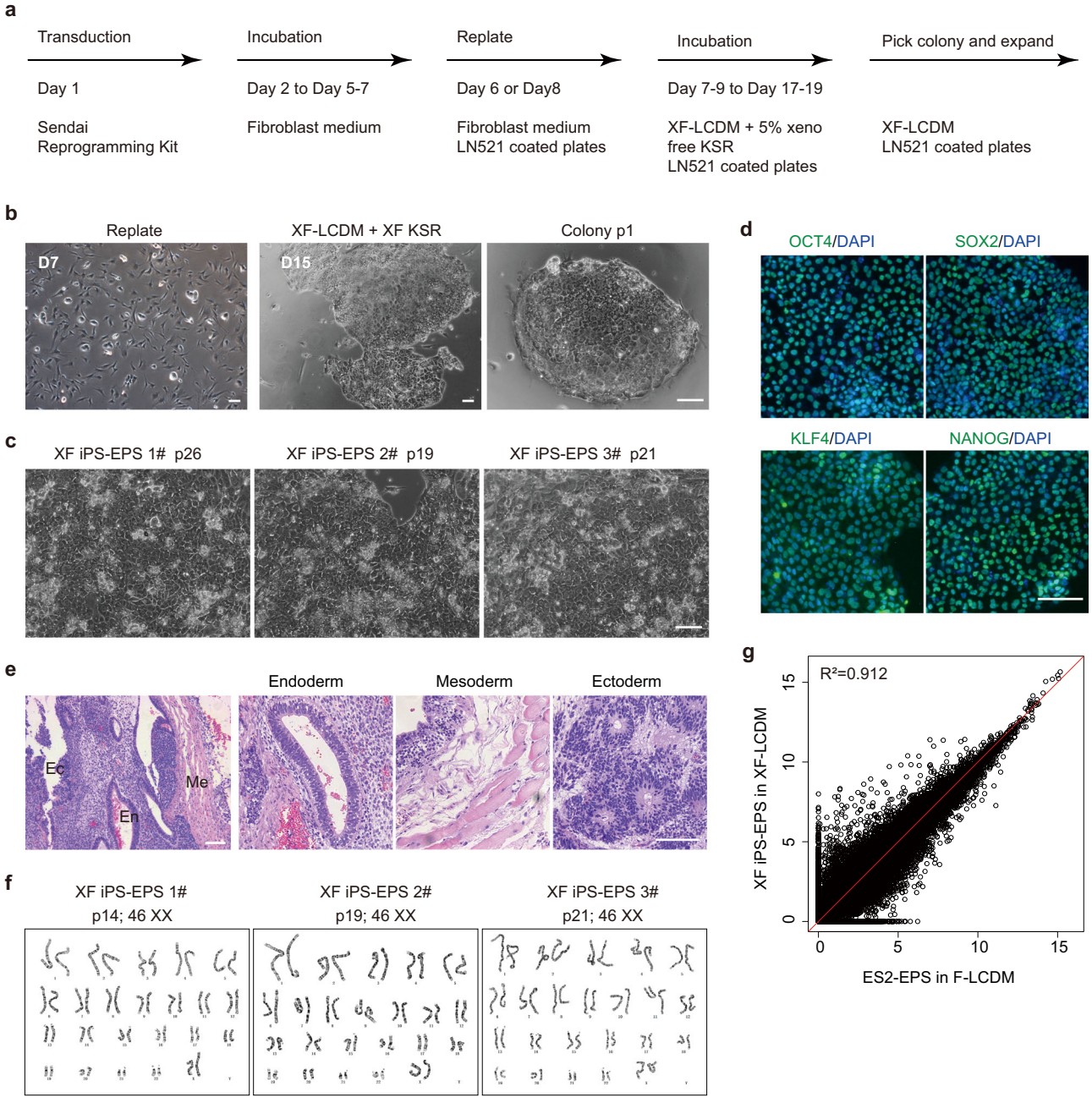

**Fig. 5 The xeno-free LCDM condition supports hEPS cell generation from fibroblasts. a** A diagram showing the procedures of generating xeno-free hEPS cells from human fibroblasts by reprogramming. **b** Representative images showing morphological changes of cells during reprogramming. Similar images were obtained in at least two independent experiments. **c** Representative images showing the morphologies of xeno-free human EPS cells derived from human fibroblasts. Similar images were obtained in at least two independent experiments. **d** Representative immunostaining images showing pluripotent marker gene expression in xeno-free hEPS cells derived from human fibroblasts. Similar images were obtained in at least three independent experiments. **e** Representative images showing histological analysis of teratomas generated from xeno-free hEPS cells derived from human fibroblasts. Similar images were obtained in at least three independent experiments. **f** Karyotype analysis of xeno-free hEPS cells derived from human fibroblasts. **g** Analysis of global gene expression correlation between feeder-cultured hEPS cells and xeno-free hEPS cells derived from human fibroblasts. F-LCDM, hEPS cells cultured on feeders; XF-LCDM, xeno-free hEPS cells. Scale bars, 100 μm. Experiments in **b**–**e** were all independently repeated at least two times with similar results.

written consent and approval by the Clinical Research Ethics Committee of China-Japan Friendship Hospital (No. 2009-50). The establishment of extended pluripotent stem cells from donated human fibroblasts was approved by the Ethics Committee of Peking University Health Science Center (IRB00001052-08093). The cross-species chimeric experiments were reviewed and approved by the Ethics Committee of Peking University Health Science Center (LA202001) and followed the ethical guidelines for human embryonic stem cell research released by the International Society for Stem Cell Research (ISSCR). Generally, human cells were injected to the 8-cell stage mouse embryos and ~15 injected embryos were transferred to each uterine horn of 0.5 or 2.5 day post-coitum pseudo-pregnant ICR females. Conceptuses were

dissected at the E6.5 or E10.5 developmental stage. Embryos, placentas and yolk sacs were isolated from conceptuses for further analysis.

**Animals**. Animal experiments were performed according to the NIH guidelines. All of the mouse experiments performed in Deng laboratory were approved by the Institutional Animal Care and Use Committee of Peking University. The mice were housed with a 12-h light/12-h dark cycle between 06:00 and 18:00 in a temperature controlled room (22 ± 1 °C) with 40–60% humidity. Water and food were accessible at all times.

**Culture of feeder-dependent and xeno-free human EPS cells**. Feeder-dependent human EPS cells were cultured in N2B27-LCDM medium with feeder cells ($3*10^4$ cells per $cm^2$). A total of 500 ml of N2B27 medium was prepared by including: 240 ml Dulbecco's modified Eagle medium (DMEM)/F12, 240 ml Neurobasal, 2.5 ml N2 supplement, 5 ml B27 supplement, 1% GlutaMAX, 1% nonessential amino acids, 0.1 mM b-mercaptoethanol. To prepare the N2B27-LCDM medium, small molecules and cytokines were added in the N2B27 medium as indicated at the following final concentrations: 10 ng/ml recombinant human LIF, 1 µM CHIR99021, 2 µM (S)-(+)-Dimethindene maleate, 2 µM Minocycline hydrochloride, 2 µM Y-27632. For some cell lines, 0.5 µM IWR-endo-1 was added.

A step-by-step protocol describing culturing and generating xeno-free hEPS cells can be found at Protocol Exchange[29]. Xeno-free human EPS (XF hEPS) cells were cultured in chemically defined medium under 20% $O_2$ and 5% $CO_2$ at 37 °C. A total of 50 ml XF hEPS medium was prepared by including: 25 ml DMEM/F12, 25 ml Neurobasal, 10 µg/ml human insulin, 5.5 µg/ml apo-Transferrin, 1 ng/ml sodium selenite, 20000X ethanolamine (liquid), 5000X human catalase (liquid), 100 µg/ml L-ascorbic acid-2-phosphate, 5–20 µg/ml Activin A, 10 ng/ml recombinant human LIF, 1 µM CHIR 99021, 2 µM (S)-(+)-Dimethindene maleate, 2 µM Minocycline hydrochloride and 5 µM Y-27632. For some of the cell lines, the addition of 0.5 µM IWR-endo-1 was recommended. If it is not necessary to make the medium be fully xeno-free, catalase from the bovine source (Sigma, C1345) can be used to replace the catalase from the human source. Prepared XF hEPS medium could be kept at 4 °C for up to 1 week.

XF hEPS cells were cultured on Laminin 521, which was diluted in DPBS with calcium and magnesium (Thermo Fisher Scientific, 2069089). The final concentration of Laminin 521 was 2.5 µg/ml. For culturing XF hEPS cells in 24-well plates, 350 µl of diluted Laminin 521 was added to one well, then the plates were incubated at 37 °C for at least 2 h or 4 °C overnight. Do not allow the culture surface to dry as the matrix will become inactivated.

XF hEPS cells were passaged at a ratio from 1:6 to 1:10 every 3–4 days, and no medium change was needed unless cultured in high density (more than $3*10^6$ cells/plate). Cell passaging was conducted when XF hEPS cells reached 85–95% of confluence. XF hEPS cells were washed with DPBS without calcium and magnesium (Thermo Fisher Scientific, 14190136) for once, then appropriate volume of 0.5X TrypLE Select (ThermoFisher Scientific, 12563029) was added and incubated for 3 min at 37 °C in the incubator. Then the TrypLE solution was aspirated and 1 ml culture medium was added. The digested cells were pipetted into single cells and seeded at desired densities. Optionally, Trypsin-EDTA (Thermo Fisher Scientific, 25300-062) or Accutase (Millipore, SCR005) could also be used instead of TrypLE Select. The specific composition of the XF hEPS cell culture medium is listed in the Supplementary Table 1.

Basic medium tested in the paper are as follows: DMEM/F12 (Thermo Fisher Scientific, 11330-032), Neurobasal (Thermo Fisher Scientific, 21103-049), DMEM basic (Thermo Fisher Scientific, C119655QOBT), Knockout DMEM (Thermo Fisher Scientific, C10829-018).

OT H1-EPS cells were converted from primed hPSC line H1, which carried an *OCT4*-Tdtomato knock-in reporter. iPS-EPS cells were generated from human fibroblasts by reprogramming. This is the first time that these cell lines were characterized and these cell lines have not been deposited in a cell bank.

**Culture of primed hPSCs and naive hPSCs**. The cell line H1 (WA01, NIHhESC-10-0043) was obtained from WiCell and authenticated by karyotype analysis. The iPS cell lines were established in our laboratory. Conventional hPSCs were cultured in mTeSR1 medium (Stem cell technology, 85850) on Matrigel (Corning, 54248) coated plates and passaged with ReleSR (Stem Cell Technology, 05873). Ten micromolar of Y-27632 was added to increase the cell viability for the first 24 h after passaging. Naive hPSCs were propagated in N2B27 supplemented with PGXL (1 µM PD0325901, 2 µM Gö6983, 2 µM XAV939 and 10 ng/ml human LIF) on irradiated MEF feeders. The converted process was completed according to a previous study[17]. For the initial passaging and culturing, Y-27632 was needed.

**Chimeric assay**. The interspecific chimeric experiments were approved by the Ethics Committee of Peking University Health Science Center (LA202001). At the time of one day before passaging, XF hEPS cells were used for chimeric experiment, which showed an optimal undifferentiated morphology and proliferated exponentially. At this time point, the colonies should be at sub-confluent density (~70% of the density on the day of passaging). Cells were digested into single cells, which were further filtered through a cell strainer (40 µm). Then, the digested cells were centrifuged at 450–600 x g at room temperature for 3 min. After removing supernatant, the cells were re-suspended in the culture medium at a proper density (2–6*10^5 cells/ml). The suspension was placed on ice before injection (no >30 min). 10–12 of the digested cells were injected into each 8-cell stage embryo. For the generation of chimeric blastocysts, the injected embryos were cultured in the KSOM medium for another 48 h before being fixed for analysis.

To generate in vivo chimeric conceptuses, chimeric embryos that were injected with XF hEPS cells were recovered for 1–2 h in a humidified incubator under 5% $CO_2$ at 37 °C and ~16 injected embryos were transferred to uterine horns of 2.5 day post-coitum pseudo-pregnant females. The conceptuses were dissected at the E6.5 developmental stage and observed using an immunofluorescence stereomicroscope

for detecting mclover+ cell localization. Naive and primed human PSCs were used as controls and the treatment method is the same as the XF-EPS cells.

**Screening of factors that can maintain hEPS cells without feeders**. To screen factors secreted by feeder cells that could maintain *OCT4* expression in hEPS cells without feeders, we used a hEPS cell line (OT H1-EPS) with an *OCT4* reporter. hEPS cells were dissociated using Trypsin-EDTA and seeded at $4*10^4$/well onto Matrigel coated 24-well plates. Cells were cultured using hEPS culture medium (N2B27-LCDM). Factors were individually added into the medium, and at least two concentrations for each factor were tested. At passage 2, cells were fixed for immunostaining and FACS analysis. Tested factors were as follows: Activin A (Stemimmune LLC, HST-A-1000), FSTL1 (Peprotech, 120-51), IGF1 (Peprotech, 100-11), IGF2 (Peprotech, 100-12), HRG1β (Peprotech, 100-03), FGF4 (Peprotech, 100-31), TGFβ1 (Peprotech, AF-100-21C), FGF2 (Peprotech, 100-18B-1000).

**Screening of matrix that can replace Matrigel in culturing XF hEPS cells**. To screen extracellular matrix that can functionally replace Matrigel, selected matrix proteins were coated according to manufacturers' suggestions. The coated plates were used immediately or sealed and stored at 4 °C for no >2 weeks. All the experiments were performed at least in 3 replications for each matrix protein, and at least two EPS cell lines were used to confirm the experimental results. hEPS cells were harvested at 1.5, 24, and 72 h after seeding, and the cell number was counted using hemocytometer. Violet staining was also performed to further confirm the results. The matrix proteins tested were as follows: Matrigel (Corning, 54248), Geltrex (Thermo Fisher Scientific, A1413202), Fibronectin (Sigma, F2006), Laminin (Sigma, L6274), Collagen (Sigma, C5533G), VTN-N (Thermo Fisher Scientific, A14700), Laminin 511 (Biolamina, LN511-0202) and Laminin 521 (Stem cell technologies, 77004).

**Adaptation of feeder-cultured hEPS cells to xeno-free hEPS cells**. hEPS cells cultured on feeders were dissociated and re-suspended in appropriate volume of N2B27-LCDM medium, and seeded on the Laminin 521-coated plates. The medium was replaced with XF hEPS culture medium 24 h later. For the first 3–5 passages, it was needed to lower the split ratio (1:3) to improve cell viability and reduce selective stress. Then cells could gradually proliferate well in XF hEPS medium. In addition, for specific cell lines that have difficulties in adaptation, 1–5% of xeno-free KSR (Thermo Fisher Scientific, 12618013) was recommended to add in the medium to increase viability and proliferation rate during the adaptation.

**Conversion of conventional hPSCs into XF hEPS cells**. To digest primed hPSCs for conversion, the conventional hPSC medium was removed from the wells, and DMEM/F12 medium was used to wash to ensure that no dead cells or cell debris remained in the culture. Then, primed hPSCs were dissociated to small clumps with ReleSR, and seeded to Laminin 521-coated plate in appropriate volume (according to the cell lines and growth ratio). Cells were cultured in mTeSR1 for the first 24 h and then changed into XF hEPS culture medium. To increase viability and proliferation rate, 1–5% of xeno-free KSR was optional, and passaging at a high density (1:3 ratio) was preferred for the first three passages. In our hand, the conversion was approximately taken about ten passages, then XF hEPS cells could be propagated well in XF hEPS medium.

**Animal component-free cryopreservation of XF hEPS cells**. XF hEPS cells were dissociated and centrifuged at 450–600 x g for 3–5 min. Then supernatant was discarded and 1 ml of cold cryopreservation solution (NutriFreez® D10 Cryopreservation Medium, Biological Industries, 05–713) was added to suspend cells, which were transferred to cryopreservation tubes. The tubes were immediately stored in Gradient cooling box at −80 °C Celsius for 24 h, then transferred to liquid nitrogen tank for long-term preservation.

**Proliferation assay**. To analyze the proliferation rate of feeder-cultured hEPS and XF hEPS cells, cells were dissociated using TrypLE and replated on feeder cells or Laminin 521 pre-coated 12-well plates respectively at a density of $8*10^4$ cells per well. Seventy-two hours later, 5 µM of EDU was added to the wells and cell were incubated at 37 °C for 2.5 h. The samples for each cell lines were analyzed in duplicate and negative controls (no EDU added wells) were also included. Then cells were harvested, washed and centrifuged. Further procedures were performed following the manufacture's standard protocol (Thermo Fisher Scientific, C10424). Flow cytometry analysis was conducted using BD FACS Verse and data analysis was performed using FlowJo software (Ashland).

**Doubling time analysis**. The assay was conducted on 24-well plates. Feeder-cultured EPS cells and XF hEPS cells were dissociated into single cells and seeded onto feeder cells or Laminin 521 pre-coated plates separately at a density of $4*10^4$ cells per well. Cell number was counted using hemocytometer every 24 h. The

doubling time was calculated using the formula:

$$DT = 48 \times \frac{\log 2}{\log Nt(\text{number of cells at day4}) - \log No(\text{number of cells at day2})}.$$

**Analysis of single-cell cloning efficiency**. To analyze the cloning efficiency of feeder-cultured and XF hEPS cells, cells were plated onto 96-well plates pre-coated with feeder cells or Laminin 521, at a density of 1, 10, 100 cells per well. For each cell line at each density, 32 duplications were performed. The colonies were counted 5 days later, and cloning efficiency was calculated as seeding cell number divided by colony number. The average efficiency at three different seeding densities were calculated.

**Immunofluorescence**. For immunofluorescence, cells were fixed in 4% paraformaldehyde (DingGuo, AR-0211) at room temperature for 15 min. Then, the fixed cells were blocked using the blocking reagent (PBS (Corning, 21-040-CVR) plus 0.25% of Triton X-100 (Sigma-Aldrich, T8787) and 3% of normal donkey serum (Jackson Immuno Research, 017-000-121)) at room temperature for 45 min. Afterwards, the cells were incubated with primary antibodies with suitable dilutions using the blocking reagent at 4 °C overnight. On the next day, primary antibodies were washed using PBS for five times (5 min of incubation in PBS for each time), and secondary antibodies (Jackson Immuno Research) were added for incubation at room temperature for 1 h. After washing out the secondary antibodies using PBS for five times, the nuclei were stained with DAPI (Roche Life Science, 0236276001). Antibody information was provided in the Supplementary Table 3.

**Flow cytometry**. Cells were harvested and fixed using BD Cytofix/Cytoperm solution (BD, 554714) for 20 min at 4 °C, and washed 3 times with PBS, stained at least 4 h with primary antibody and 1 h with secondary antibody. Then cells were washed again and analyzed using BD FACS Verse. Data analysis was performed using FlowJo software. All the experiments were conducted in duplicate for each treatment. Antibody information was provided in the Supplementary Table 3.

**Analysis of Neu5Gc expression**. Analysis of Neu5Gc expression was performed to demonstrate the absence of animal components in xeno-free culturing system. The analysis procedure was based on previous studies[30,31]. Briefly, xeno-free hEPS cells and hEPS cells cultured on different conditions were harvested using TrypLE and washed once with 1 ml cold PBS, which were gently centrifuged at 4 °C. The following steps were performed according to the instructions of Anti-Neu5Gc Antibody Kit (Biolegend, 146901). The treated samples were analyzed using BD FACS Verse. Data analysis was performed using FlowJo software. All the experiments were conducted in duplicate for each treatment.

**EB formation assay**. XF hEPS cells were dissociated into single cells and cultured for 7 days on ultra-low attachment 6-well plates in IMDM (Thermo Fisher Scientific, 12440-053) supplemented with 15% FBS (VISTECH, VIS8618005) at a density of $5*10^5$ per well. To increase cell viability, 5 μM of Y27632 was added to the medium for the first 24 h. Medium change was performed every 3 days. Then, EBs were collected and plated on the Matrigel coated 24-well plates for another 7 days in the same medium. Then cells were fixed for analysis. Antibody information was provided in Supplementary Table 3.

**Teratoma assay**. Approximately $5*10^6$ of XF hEPS cells were suspended in 50 μl XF hEPS medium, and mixed with the same volume of Matrigel (thawed before the experiment on ice). The cell mixture was sub-cutaneously injected into immuno-deficient NPG mice. Within 4–8 weeks, teratomas were developed. The mice were killed before the tumor size exceeded 1.5 cm in diameter. The teratomas were isolated and embedded in paraffin, which were processed for hematoxylin and eosin staining.

**Karyotype analysis**. Cell were collected at a density of 60 - 80% of confluence on the day of sampling. After 2 h of incubation with fresh medium, Colcemid solution was added to the culture at a final concentration of 0.02 mg/ml. Then the cells were incubated for 1 h. After incubation, cells were washed, digested and centrifuged. To obtain a single-cell suspension, the pellet was re-suspended in hypotonic solution (0.56% KCl), and incubated at room temperature for 6 min. After centrifuging and removing the hypotonic solution, 5 ml of ice-cold fixative (3:1 methanol: acetic acid) was added to the suspension in a dropwise manner. Then the cells were incubated at room temperature for 5 min before spinning down. The fixing procedure was repeated for additional three times. Afterwards, the pellet was re-suspended in a final volume of 1 ml fixative. Then, the cells were dropped onto 5% acetic acid ± ethanol (ice-cold) washed slides and stained with Giemsa. For each experiment, 20 metaphases were analyzed. The number of chromosomes and the presence of structural chromosomal abnormalities were examined.

**Trophoblast differentiation and derivation of trophoblast stem-like-cells from XF-EPS cells**. XF hEPS cells were dissociated and seeded into 24-well Matrigel

coated plates at a density of $2.5*10^4$ cells per well, and cells were cultured in XF hEPS medium for the first 24 h. To initiate differentiation, the culture medium was changed on the second day with: 40 ml DMEM/F12 and 40 ml IMDM (Thermo Fisher Scientific, 12440-053) with 20% KSR (Thermo Fisher Scientific, A3181502), 10 ng/ml BMP4, 1 μM A83-01 (Tocris, 2939), 0.1 μM PD173074 (Selleck, S1264). The differentiated cells were harvested for analysis from day 1 to day 8. mRNAs were extract using Trizol and FACS samples were prepared using BD Cytofix/Cytoperm Kit (BD, 554714).

Conventional hPSCs and XF-hEPS were dissociated by ACCUTASE or TrypLE Express into single cells, about $5*10^5$ cells were seeded on a 6-well plate pre-coated with MEF feeder cells and cultured in 5% $CO_2$ and 20% $O_2$ incubator. Cells were cultured in the initial medium for the first 24 h and then changed into TSC medium, which consist of DMEM/F12 supplemented with 0.1 mM 2-mercaptoethanol, 0.2% FBS, 0.5% penicillin-streptomycin, 0.3% Bovine Serum Albumin, 1% Insulin-Transferrin-Selenium-Ethanolamine supplement, 1.5 μg/ml L-ascorbic acid, 50 ng/ml hEGF, 2 μM CHIR99021, 0.5 μM A83-01, 1 μM SB431542, 0.8 mM valproic acid and 5 μM Y-27632. hTS-like cells could be passaged using ACCUTASE or TrypLE every 4 days at a 1:3 to 1:4 split ratio. Media was changed every day.

**Evaluation of OCT4 DE transcriptional regulation**. To evaluate human OCT4 DE transcriptional regulation in primed hPSCs and hEPS cells cultured in different conditions, OCT4-DE luciferase plasmid (Addgene, 52414) was transfected into the cells by nucleofection (4D-Nucleofector™ System, Lonza). A control vector pGL4.74 [hRluc/TK] (Promega, E6921) was co-transfected for normalization. After transfection, feeder-cultured EPS cells were seeded onto Matrigel coated 96-well plates at a density of 5,000 cells per well and XF hEPS cells were seeded onto Laminin 521-coated 96-well plates in xeno-free hEPS medium. Primed PSCs were used as controls. After 3 days, the cells were lysed for detecting the luciferase activity using the Dual-Luciferase Reporter Assay System (Promega, E1960) and analyzed using Cytation 5. For each cell line, three replicates were included. Baseline activity was analyzed by transfection with an empty vector.

**Detection of XF hEPS derivatives in mouse embryo**. The recovered embryos were washed with PBS containing 3 mg/ml PVP (Sigma, P0930) for two times and fixed with 4% PFA at room temperature for 15 min. Then the fixed embryos were washed again with PBS/PVP for three times. Permeabilization was conducted with PBS/PVP plus 0.25% TritonX-100 at room temperature for 2 h. After that, the embryos were blocked with the blocking buffer (PBS/PVP + 0.1% BSA + 0.01% Tween-20 (Sigma, P1379) + 2% Donkey Serum) at room temperature for 1 h. For immunostaining, embryos were stained with anti-GFP, CDX2, CK18, CK8, GTAT6 and OCT4 antibodies at the same time at 4 °C overnight. Then embryos were washed with blocking buffer three times and each time lasted for 15 min. Secondary antibodies were stained at room temperature for 1 h. The nuclei of embryos were stained by DAPI at room temperature for 10–15 min. Then the embryos were washed with PBS/PVP for three times. All these samples were imaged by the Dragonfly (Andor). Antibody information was provided in the Supplementary Table 3.

**Human fetal fibroblast (HFF) culture and the establishment of XF hEPS cells from HFFs**. HFFs were isolated from skin and further cultured in fibroblast culture medium, which contained DMEM (Thermo Fisher Scientific, C119655QOBT) plus 10% fetal bovine serum (Thermo Fisher Scientific, 12483020) and 0.5% penicillin and streptomycin (Thermo Fisher Scientific, 15140-122).

To establish XF hEPS cell lines, HFFs were seeded at $1*10^5/cm^2$ in 6-well plate and further cultured for 1–2 days in fibroblast culture medium. The CytoTuneTM-iPS 2.0 Sendai Reprogramming Kit (Thermo Fisher Scientific, A16517) was applied to reprogram HFFs. Sendai virus transduction was performed according to the user's manual. 5–7 days post transduction, infected HFFs were replated onto Laminin 521-coated 6-well plate at $1*10^5/cm^2$. After overnight culturing in fibroblast culture medium, cultured medium was refreshed to xeno-free hEPS culture medium plus 5% xeno-free KSR. From 14 days post transduction and onwards, hEPS-like colonies emerged and could be picked up. Colonies were enzymatically dissociated into single-cell using TrypLE Select and after 4–7 days colonies could be observed. In order to improve the cell survival rate, it was recommended that mechanical digestion was used to passage the cells during the first three passages. After about 3–5 passages, EPS cell lines can be stably maintained in XF hEPS culture medium without KSR.

**Quantitative PCR**. The total RNAs were isolated using the Direct-zol RNA Kits (ZYMO Research, R2051). RNA was converted to cDNA using TransScript First-Strand cDNA Synthesis SuperMix (TransGen Biotech, AT301). Quantitative PCR analysis was conducted using the KAPA SYBR FAST qPCR Kit (KAPA Biosystems, KK4601) with the Bio RAD CFX Connect Real-Time System. The primers that were used for Q-PCR analysis are listed in Supplementary Table 4. The data were analyzed using the delta-delta CT method.

**Gene targeting in xeno-free hEPS cells**. Guide RNA sequences were cloned into the plasmid px330 (Addgene, 42230). Px330 containing gRNAs were co-transfected

with the targeting vector into digested single xeno-free hEPS cells by nucleofection (4D-Nucleofector™ System, Lonza). mclover+ cells were purified using FACS, and further seeded as single cells. Single colonies were picked and expanded individually. Genomic DNA of colonies were extracted using the DNeasy Blood & Tissue Kit (Qiagen), which was further analyzed by genomic PCR.

**Human mitochondrial PCR assay.** Total DNA was isolated using the DNeasy Blood & Tissue Kit (QIAGEN, 69506). For detecting human-specific mitochondrial DNA element by Q-PCR, 18 ng of total DNA per sample was used. The data was analyzed using the delta-delta CT method, which was first normalized to the values of human-mouse conserved mitochondrial DNA element. Then the relative expression values were further normalized to the values generated from control samples isolated from un-injected wild-type mouse tissues. The primers that were used for Q-PCR analysis are listed in Supplementary Table 4.

**RNA fluorescent in situ hybridization (FISH).** To detect the X chromosomal state of xeno-free hEPS cells, commercial *XIST* probes (Biosearch Technologies, SMF-2038-1) were used. XF-hEPS cells and naive human PSCs were seeded onto μ-Slide 8-well (Ibidi, 80826) plates at 4–5*10⁴ per well two days before the analysis. Cell were washed with 200 μl PBS and fixed with 4% paraformaldehyde for 10 min at room temperature, and then permeabilized with 70% Ethanol for at least 1 h at 4 °C. Then cells were washed with Wash buffer A for 5 min and hybridized with labeled probes 1:50 diluted with hybridization buffer at 37 °C for 12–16 h. After that, cells were washed with Wash buffer A once again at 37 °C for 30 min and protected from light. Cells were further stained with DAPI for 1–2 min and washed with Wash buffer B for 2–5 min. Finally, the plates were sealed with mounting medium H-1000 (Vector Laboratories, H-1000) and visualized with High Speed Spinning Disk Confocal Microscope.

**Statistical analysis.** All values are depicted as mean ± SD. Statistical parameters including statistical analysis, statistical significance, and $n$ value are reported in the Figure legends and Supplementary Figure legends. Statistical analyses were performed using Prism Software (GraphPad). For statistical comparison of multi-groups, we performed one-way ANOVA analysis with Tukey multiple comparison test. A value of $P < 0.05$ was considered significant.

**Transcriptome analysis.** Total RNA was isolated from feeder-cultured and xeno-free EPS cells using the RNeasy Mini Kit (Qiagen, 74106). RNA sequencing libraries were constructed by using magnetic beads with oligo (DT) enriched mRNA. After that, fragment buffer was added to break the mRNA into short segments. Using mRNA as template, a strand of cDNA was synthesized using six base random primers. Then buffer, dNTPs, DNA polymer I and RNase H were added to synthesize a two strand cDNAs, which were purified using AMPURE XP beads. The purified double stranded cDNAs were repaired, a-tailed and sequenced, and then the fragment size was selected by AMPURE XP beads. Finally, PCR amplification was carried out and PCR products were purified with AMPURE XP beads to obtain the final library. The fragmented and randomly primed $2 \times 100$-bp paired-end libraries were sequenced using an Illumina HiSeq 2500. The generated sequencing reads were mapped against human genome build hg19 for human using TopHat alignment software tools. The read counts for each gene were calculated, and the expression values of each gene were normalized using FPKM. Scatter plots were generated in R v3.6.1 using the graphics package. Differentially expressed (DE) genes were identified using the package DESeq2 in the R software. The threshold of being significant for declaring gene expression differences include: an adjusted $P$-value $< 0.05$ and an absolute value of the log2 ratio $> 1$. The identified DE genes were used for gene ontology analysis by subjecting the gene lists to clusterProfiler bioinformatics tool[32].

**GSEA analysis.** To perform GSEA analysis, gene sets were first selected, including Module C and D of EPS from our previous paper (Yang et al.[6]), markers of pre-implantation EPI in E5/E6/E7 (Petropoulos et al.[19]), and Naive signatures (Liu et al.[35]). Some public datasets were also collected, including datasets of pre-implantation embryos (GSE36552, Yan et al.[20]) and two naive cell datasets (E-MTAB-4461, Guo et al.[21], 2016; E-MTAB-5674, Guo et al.[17]). All datasets were preprocessed using the same bulk RNA-seq analysis pipeline as described above. The normalized FPKM difference between non-primed samples and the mean of primed samples were used as input of GSEA. For embryo datasets, the expression values of all cells in the same cell type were averaged at first. The Normalized Enrichment Score (NES) of all analyzes were calculated using R package clusterProfiler with 10,000 permutations per run. To visualize the contrast clearer, for primed samples of each dataset, the normalized FPKM difference of primed samples and the mean of other non-primed samples were used as input of GSEA. The results of multiple GSEA analyzes were visualized in form of heatmap using R package ComplexHeatmap[33].

**Single-cell RNA-seq analysis.** Single-cell RNA-seq data were preprocessed using cellranger-3.1.0. The output gene expression matrix was analyzed using R version 3.6.1, and the R packages used below were mostly available in Bioconductor version

3.10. We loaded data into R environment using package DropletUtils. Low-quality cells with low log total UMI counts(<median – 3*median absolute deviation), low log total genes(<median – 3*median absolute deviation) and high mitochondrial gene proportions (>median + 2*median absolute deviation) were filtered using package scatter. Then we normalized UMI counts to ln(TPM + 1). The normalized data were clustered and performed differentially expressed gene analysis using R package Seurat3. The outlier clusters with significantly low UMI counts and gene numbers were removed.

To compare our EPS single-cell data with published single-cell data of naive and primed cells (Liu et al.[35]), we integrated our dataset to dataset GSE150311 and GSE147564 using integration method of Seurat3. Then the gene signature scores of collected markers of pre-implantation EPI/PE/TE in E5/E6/E7 (Petropoulos et al.[19]), markers of ICM, pre- and post- implantation EPI and post- implantation PE/TE (Xiang et al.[16]) and Naive signatures (Liu et al.[35]) were calculated using the same method descried in the original paper respectively. The mean values of all signatures of all cell types were visualized in the form of Heatmap using R package ComplexHeatmap[34]. The signature scores of cell types were also used to visualize in violin plot.

**ATAC-sequencing analysis.** Cells were prepared by using the TruePrep DNA Library Prep Kit V2 for Illumina (Vazyme, TD502-01) following the manufacture's guideless. The purified DNA libraries were sequenced using an Illumina HiSeq 2500. The ATAC-seq reads were trimmed using Trimmomatic-0.39 to remove sequencing adapters. The clean reads were next mapped to human reference genome hg19 using bowtie2 version 2.3.5 with parameter "very sensitive". Low quality mapped reads were removed using samtools version 1.10. The peaks were called using MACS2-2.2.6 with parameters "-g hs -nomodel -shift -100 -extsize 200". The differential binding analysis of peak data was performed using R package DiffBind. The differentially binding peaks (FDR < 0.05) between groups were used to perform motif enrichment analysis using Homer version 4.10.3. The genes whose transcription start sites within 10 kb around differentially binding peaks were collected to perform GO analysis using R package clusterProfiler. The peaks within mitochondrial region were not included in all analysis above.

The filtered.bam files before peak calling were converted to.bw files using tool bamCoverage of deepTools-3.5.0 with parameter "-normalizeUsing RPKM". These .bw files were used to calculate the signal density of given region (2 kb up- and down-streams of TSS of pre- implantation TE markers (Petropoulos et al.[19]) and randomly selected genome regions of the same length) using tools computeMatrix and plotHeatmap of deepTools.

**Whole-genome sequencing (WGS) analysis.** We performed quality control to raw WGS reads using Fastqc and Trimmomatic-0.39. The trimmed clean reads were mapped to human reference genome hg19 using bwa mem algorithm. The mapped.sam files were converted to.bam files and sorted using samtools. Then deduplication was performed using MarkDuplicates program of Picard-tools. We followed the workflows of Genome Analysis Toolkit version 4 (GATK4) to perform base recalibration, call candidate variants, learn orientation bias artifacts and filter variants one after another using involved tools of GATK4. All variants passed the filter were extracted from.vcf files using vcftools. We further filtered the variants by removing variants with approximate read depth less than 30 to get high-quality variants. We annotated all variants using Ensembl Variant Effect Predictor (VEP) web interface[33,34]. Then we analyzed VEP output files and counted numbers of variants located in different genomic regions using R. In addition, we performed prediction of copy-number alterations and loss of heterozygosity using Control-FREEC with recommended settings. The outputs were also visualized using R.

**Reporting summary.** Further information on research design is available in the Nature Research Reporting Summary linked to this article.

## Data availability
The accession number of the sequencing data reported in this paper is NCBI GEO: GSE156920 (single-cell RNA-seq), GSE147839 (bulk RNA-seq) and GSE157237 (ATAC-seq), the WGS data number is SRP282164. Source data are provided with this paper.

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

## Acknowledgements

We thank X. Zhang and Q.Z. Feng for technical assistance. We thank J.Y. Guan, Y. Fu, and Y.Q. Li for discussions in the course of the preparation of this manuscript. We thank the flow cytometry and confocal microscopy Core at National Center for Protein Sciences at Peking University, particularly H.X. Lyu and H.Y. for technical help. This work was supported by National Key Research and Development Program of China (2017YFA0103000, 2018YFA0108102), the National Natural Science Foundation of China (31521004, 31730059), the Beijing Science and Technology Major Project (Z191100001519001) and the grants from the Beijing Municipal Science and Technology Commission (D171100000517004).

## Author contributions

B.L., S.C., Y.X.X., J.X., and H.K.D designed the study, performed, and interpreted experiments. Y.L. performed the sequencing data analysis. J.L.W. and C.L. helped with RNA-seq data analysis. Y.Y.D. helped with the reprogramming experiments. Y.C.S. helped with the experiments. H.M.L., H.Y.Z., and M.Y. helped with microinjections and mouse breeding. W.F.L. and A.Q.X. assisted in teratoma experiments. H.K.D. conceived and supervised this project and wrote the manuscript with B.L., J.X., J.L.W., and Y.B.

## Competing interests

The authors declare no competing interests.
