## [Peer Review File · Nature Communications]

Reviewers' Comments:

Reviewer #1:

Remarks to the Author:

Liu et al. have improved their manuscript in response to the reviewers' concerns. In particular, they have clarified that xeno-free EPSC have reduced chimera potential in E10.5 mouse-human chimeras compared to feeder-dependent EPSC and have removed the experiments using BCL2 overexpression. They have also performed additional experiments demonstrating that XF-EPSC can differentiate into human trophoblast stem cells, which was previously reported using naïve cells and feeder-dependent EPSC.

While I'm mostly satisfied with the revised manuscript, the authors' interpretation of the developmental stage of xeno-free EPSC remains problematic. In response to questions from two of the reviewers, they have removed the single cell RNA-seq analysis that showed XF-EPSC overlapping with various timepoints ranging from embryonic days E7-E12. Instead, they have added a GSEA analysis, which shows enrichment of "E5/6/7 epiblast enriched genes" defined by Petropoulos et al. This analysis indicates that XF-EPSC more closely resemble pre-implantation EPI compared to PE or TE, but does not take into account post-implantation timepoints. Such an analysis is included in the Supplement (Fig. S5d), but this panel clearly shows that XF-EPSC correlate more weakly with pre-EPI and more strongly with post-EPI when compared to naïve hPSC. Therefore, the correct interpretation is that EPSC represent a unique state that is distinct from both naïve and primed cells. The authors should state this point more clearly in the Results and Discussion as it currently reads as if EPSC exhibit all molecular features of naïve pluripotency (see lines 124 and 226).

Other points:

1. Suppl. Fig. 4d: Based on the image provided it would be more prudent to state that XF-LCDM cells exhibit a range of different XCI states with some cells showing no XIST, monoallelic XIST, or biallelic XIST expression.
2. The Supplementary Methods section and Fig. S5d refer to a study "Lifeng et al., 2020" which should instead refer to "Xiang et al., 2020".

Reviewer #2:

Remarks to the Author:

In their revision, the authors have performed some additional data analyses and have changed some of the claims in the previous version to reflect more accurately the experimental results. The precision of the manuscript has been improved, though I still question precisely what the EPS cells represent in embryological terms. The data as presented do not directly address heterogeneity within EPS cultures (clearly heterogeneity is present based on X inactivation data in Figure 4d) and addressing this issue might clear up some of the uncertainty. The authors might consider analysis of heterogeneity in the cultures to be beyond the scope of this study, but they should acknowledge this possibility in their discussion.

REVIEWER COMMENTS

Reviewer #1 (Remarks to the Author):

Liu et al. have improved their manuscript in response to the reviewers' concerns. In particular, they have clarified that xeno-free EPSC have reduced chimera potential in E10.5 mouse-human chimeras compared to feeder-dependent EPSC and have removed the experiments using BCL2 overexpression. They have also performed additional experiments demonstrating that XF-EPSC can differentiate into human trophoblast stem cells, which was previously reported using naïve cells and feeder-dependent EPSC.

While I'm mostly satisfied with the revised manuscript, the authors' interpretation of the developmental stage of xeno-free EPSC remains problematic. In response to questions from two of the reviewers, they have removed the single cell RNA-seq analysis that showed XF-EPSC overlapping with various timepoints ranging from embryonic days E7-E12. Instead, they have added a GSEA analysis, which shows enrichment of "E5/6/7 epiblast enriched genes" defined by Petropoulos et al. This analysis indicates that XF-EPSC more closely resemble pre-implantation EPI compared to PE or TE, but does not take into account post-implantation timepoints. Such an analysis is included in the Supplement (Fig. S5d), but this panel clearly shows that XF-EPSC correlate more weakly with pre-EPI and more strongly with post-EPI when compared to naïve hPSC. Therefore, the correct interpretation is that EPSC represent a unique state that is distinct from both naïve and primed cells. The authors should state this point more clearly in the Results and Discussion as it currently reads as if EPSC exhibit all molecular features of naïve pluripotency (see lines 124 and 226).

[Response]: According to the reviewer's comments, we have revised our manuscript to state the conclusion more clearly that EPS cells represent a unique state that is distinct from naïve and primed hPSCs (page 4: line 125-134, page 6: line 227-231).

Other points:

1. Suppl. Fig. 4d: Based on the image provided it would be more prudent to state that XF-LCDM cells exhibit a range of different XCI states with some cells showing no XIST, monoallelic XIST, or biallelic XIST expression.

[Response]: According to the reviewer's comment, we have revised related descriptions in the revised manuscript (page 3: line 102-105).

2. The Supplementary Methods section and Fig. S5d refer to a study "Lifeng et al., 2020" which should instead refer to "Xiang et al., 2020".

[Response]: We have corrected related reference in the revised manuscript.

Reviewer #2 (Remarks to the Author):

In their revision, the authors have performed some additional data analyses and have changed some of the claims in the previous version to reflect more accurately the experimental results. The

precision of the manuscript has been improved, though I still question precisely what the EPS cells represent in embryological terms. The data as presented do not directly address heterogeneity within EPS cultures (clearly heterogeneity is present based on X inactivation data in Figure 4d) and addressing this issue might clear up some of the uncertainty. The authors might consider analysis of heterogeneity in the cultures to be beyond the scope of this study, but they should acknowledge this possibility in their discussion.

[Response]: We agree with the reviewer that future studies are required to clarify the developmental identity of hEPS cells, which has been stated in the revised manuscript (page 7: line 239-241). According to the reviewer's suggestions, we have performed a preliminary analysis using the single cell sequencing data, and found that no obvious heterogenous expression of pre- or post-implantation epiblast markers were observed in hEPS cell populations at the single cell level (Supplementary Fig. 5e). Further single cell multi-omics analysis is required to understand the heterogeneity of hEPS cell cultures, which is important for clarifying the developmental identity of hEPS cells. We have added related discussion in the revised manuscript (page 7: line 240-248).